# Intersectoral and multisectoral approaches to enable recovery for people with severe mental illness in low- and middle-income countries: A scoping review

recovery; severe mental illness; community-based initiatives; collaborative care; developing countries

**Author for correspondence:**
André J. van Rensburg,
Email: jansevanrensburga@ukzn.ac.za

André J. van Rensburg[1] 🆔 and Carrie Brooke-Sumner[2] 🆔

[1]Centre for Rural Health, University of KwaZulu-Natal, Durban, South Africa and [2]Alcohol, Tobacco and Other Drug Research Unit, South African Medical Research Council, Cape Town, South Africa

## Abstract

The needs of people with severe mental illness are complex and require a range of services embedded in well-coordinated systems of care to enable recovery, promote well-being and optimise social integration. The concept of recovery is strongly rooted in the centrality of multi and intersectoral systems of care, and, while multi and -intersectoral dimensions of mental health systems have been highlighted in analyses focusing on high-income regions, little has been elaborated in terms of these approaches in the recovery of people with severe mental illness (SMI) in low- and middle-income countries (LMICs). The aim of this review was to identify and describe multi and intersectoral approaches underpinning community-based SMI recovery interventions in LMICs. A scoping review was carried out following the following steps: (1) Objectives for the review were developed and refined; (2) A systematic search of databases (EbscoHost, PubMed, Google Scholar) and previous reviews were undertaken from 2012 to 2022, where relevant papers were identified; (3) Papers with a focus on SMI and recovery, a specific description of an intervention, located in LMICs, with explicit linkages between sectors, and published in English, were selected for inclusion; (4) Data were extracted and charted and (5) Findings were analysed and reported thematically. Thirty-six papers were included for analysis, from 18 countries, including qualitative studies, trials, desktop and secondary data reviews and case studies. Examples of multi- and intersectoral action included collaboration between healthcare and community support systems, collaboration in providing supported housing and supportive community spaces for recovery, and linkages between biomedical and social spheres of care. Barriers included the dominance of mental health professions in delivering care, community-based stigmatising attitudes towards SMI. Multi- and intersectoral collaboration for SMI recovery requires investments in financing, education and coordination by a governing body.

## Impact statement

Despite a large body of work on recovery for people living with severe mental illness (SMI), and its implicit embeddedness in collaboration across sectors, little systematic description has been undertaken of its implementation in low- and middle-income countries. Our review fills this gap by providing a synopsis of how multi- and intersectoral collaboration in supporting recovery occur in these contexts. It highlights examples that involve collaboration between healthcare and community support systems, collaboration in providing supported housing and supportive community spaces for recovery, and linkages between biomedical and social spheres of care. There are, however, barriers to collaborating across sectors, including the dominance of mental health professions in delivering care, community-based stigmatising attitudes towards SMI, and a discomfort of some healthcare workers to work beyond the professional boundaries of healthcare. Multi- and intersectoral collaboration for SMI recovery needs to be driven by formal structures and financing, including both on macro and micro levels of engagement.





## Introduction

People living with severe mental illness (SMI) have substantially increased relative mortality risk compared to the general population, related to cardiovascular disease (Ali et al., 2022; Lambert et al., 2022), and in low- and middle-income countries (LMICs) particularly related to poverty that leads to poor health status (e.g., undernutrition) (Jenkins et al., 2011c; Tirfessa et al., 2019). While symptoms of the illness play a role in course and outcomes, globally and in LMIC particularly, people with SMI may experience social and economic adversities and human rights

abuses that can create a social environment that hampers clinical and personal recovery (Brooke-Sumner et al., 2014; Patel, 2016; Asher et al., 2017). Recovery, as it has been conceptualised in high-income country (HIC) settings, is described as an individual journey of transformation and personal growth moving from the distress of the acute experience of the condition towards finding meaning and purpose, a sense of belonging, forming or rebuilding meaningful relationships (Frost et al., 2017), bringing hope, empowerment, goal orientation and fulfilment (Warner, 2009; Drake and Whitley, 2014; Whitley et al., 2015). Recovery encompasses concepts of prosperity (legal, political and economic dimensions); individual recovery (dimensions of normalcy, knowledge, individuality, responsibility and identity); clinical recovery (treatment and diagnosis dimensions) and social recovery (externally and internally derived notions of social awareness, being a part of society, functioning well within, groups, treated as an equal) (Vera San Juan et al., 2021). Biomedically oriented health systems alone are inadequately configured to address the spectrum of these recovery needs which extend across intersecting social, economic, cultural and political spheres, beyond the health sector (Gamieldien et al., 2022). While many of the recovery concepts may be cross-cutting among HIC and LMIC, some concepts, developed in Western sociocultural contexts, may be limited in being rooted in economic environments and health and social welfare systems able to provide for people's material needs (Gamieldien et al., 2021). In LMICs there may be greater involvement of families in providing care and supportive environment, use of non-Western healing approaches (Onken et al., 2007), and a more important role of spirituality in recovery (Gamieldien et al., 2021).

Since its introduction into health policy discourse in the 1970s, "intersectoral action" has become a staple in framing responses to public health challenges. The need for the health sector to collaborate with a range of other sectors to improve health outcomes continues to be highlighted (Sanni et al., 2019). More recent conceptualisations include "multisectoral action for health" which refers to the deliberate or collateral inclusion of different actors and sectors in health improvement, including initiatives such as "Whole of Government," "Joined-up Government" approaches, horizontal and integrated policymaking, and Health in All Policies. Despite the conceptual promise of inter-and multisectorality, and evidence of its implementation in HIC (Diminic et al., 2015; Jørgensen et al., 2020; Jørgensen et al., 2021; Mondal et al., 2021) this has not consistently translated into policy or services. For instance, neither the WHO's Innovative Care for Chronic Conditions Framework (Nuño et al., 2012), nor its subsequent modification for LMICs or countries in health transition (Oni et al., 2014), adequately considers the role of sectors outside of health. A well-documented example of the costs of failure to approach community mental health from an intersectoral approach is the US deinstitutionalisation movement. Following the policy shifts towards deinstitutionalisation, financial costs and responsibilities were dispersed through various stakeholders and agencies. This led to a fractured system, inadequate to address the complex needs of people with SMI, leading to homelessness or incarceration when placed in community settings (Grazier et al., 2005). In order to develop more people-centred, humane and effective community mental health systems, recovery should be firmly couched in service and strategic collaboration across sectors (Drake and Whitley, 2014). Several examples of promising shifts towards intersectoral collaboration in community SMI services have emerged in high-income settings. Intersectoral service networks in Belgium (Nicaise et al., 2021) and Canada (Fleury et al., 2017) includes integrated,

intersectoral collaboration in the form of housing, educational and employment support, beyond medical and psychiatric care. The Australian Partners in Recovery model is a good example of how care coordination can aid recovery for people living with SMI (Isaacs, 2022). A review of interventions that focus on system-level intersectoral linkages involving mental health services and non-clinical support services yielded 40 examples from HICs, with various different collaboration modalities. Outcomes reported were largely positive, particularly regarding improved interagency communication, mutual understanding and empathy, cost efficiency, involvement of lay health workers, as well as various service user outcomes such as clinical functioning, employment prospects and accommodation stability (Whiteford et al., 2014). This being noted, the connection between recovery and intersectoral care remains relatively ill-defined and several gaps remain in this body of evidence (Jørgensen et al., 2021). However the relevance and need for development of this approach to recovery services are highlighted in the 2022 World Mental Health report (World Health Organization, 2022).

While there is much promise of inter- and multisectoral approaches to SMI recovery, there is paucity of reviews on the subject – particularly in LMICs, and a lack of systematised evidence on how to implement the approach. While the implementation of intersectoral collaborations to enable recovery of people living with SMIs have been well-described in HICs, it remains uncertain how intersectoral care is being pursued in contexts faced with a lack of resources and infrastructure, mental health system investment-to-population ratio, substantial geographical and cultural variation and underdeveloped welfare systems (Patel, 2016). The aim of this scoping review was therefore to identify and describe multi and intersectoral approaches underpinning community-based SMI recovery interventions in LMICs.

## Methods

This scoping review was guided by the methodological steps outlined by Arksey and O'Malley (2005) and the Johanna Briggs Institute (JBI, 2015), following the following phases: (1) Objectives for the review were developed and refined among the authors, based on a brief, initial literature review; (2) A systematic search of databases was undertaken where relevant papers were identified; (3) Relevant papers were selected for inclusion; (4) Data were extracted from these selected studies, and were charted according to the Preferred Reporting Items for Systematic reviews and Meta-Analyses Extension for Scoping Reviews (PRISMA-ScR) (Tricco et al., 2018) and (5) Findings were thematically analysed and reported.

The search was undertaken by the authors, with weekly discussions to compare results and discuss inclusions and exclusions. We applied search terms as used in a recent scoping review exploring recovery of people living with SMI in LMICs (Gamieldien et al., 2021), with updated time parameters to reflect our search scope of 2012–2022. This resulted in an additional 12 papers added to their results (22 in total). We then conducted searches using key terms related to recovery, SMI, community settings, LMICs (see Supplementary Material for a full description of search terms), in EbscoHost (Academic Search Complete; APA PsycInfo; Health Source – Consumer Edition; Health Source: Nursing/Academic Edition; MasterFILE Premier; MEDLINE with Full Text), PubMed and Google Scholar. In Google Scholar, the terms and related terms "recovery," "SMI," "community settings," and "LMICs" were

**Table 1.** Inclusion and exclusion criteria

| Inclusion criteria | Exclusion criteria |
|---|---|
| A primary focus on people living with SMI, as defined by the National Institute for Mental Health ("severe" and "serious" were used interchangeably), that is, "…a mental, behavioural, or emotional disorder resulting in serious functional impairment, which substantially interferes with or limits one or more major life activities. The burden of mental illnesses is particularly concentrated among those who experience disability due to SMI." (National Institute of Mental Health, 2022). This includes specific ICD-10 diagnostic categories F20–29, F30–39, F30.2, F31.2, F31.5, F32.3, F33.3 | Primary focus not on people living with SMI |
| A description of a specific intervention, programme or service | General overviews or descriptions of health systems or services, rather than a specific focus on an intervention aimed at recovery from SMI |
| A focus on community-based, outpatient settings | A focus on inpatient, institutionalised settings |
| A focus on the enabling of recovery as defined in the Introduction | No focus on dimensions of recovery |
| Adults (aged 18 years and above) | People under 18 years of age |
| Studies reported during the past decade (2012–2022) | Studies reported before 2012 |
| Studies with a primary location in LMICs (World Bank, 2022) | Studies focusing on settings in HICs (World Bank, 2022) |
| Explicit description of intended linkages with sectors other than health | No apparent linkages of an intervention with sectors beyond health |
| Published in English | Full text not in English |

included and results were screened until two sequential pages did not yield any further papers that adheres to the inclusion criteria. During the screening and review process, it became apparent that the interchangeable and ambiguous application of complex terms such as recovery, multi-and intersectoral approaches, may limit the number of papers found in databases. Therefore, an additional review of the results of 11 systematic reviews on psychosocial interventions with a focus on SMI was undertaken (Brooke-Sumner et al., 2015; Lutgens et al., 2017; Sin and Spain, 2017; Davies et al., 2018; Frederick and VanderWeele, 2019; Alhadidi et al., 2020; Al-Sawafi et al., 2020; Bighelli et al., 2021; Morillo et al., 2022; Rodolico et al., 2022; Solmi et al., 2022), while peer reviewers helpfully pointed out additional omissions in the results. This underlines the importance of including an additional consultation phase in scoping reviews, framed as an optional step in existing guidelines (Levac et al., 2010). Inclusion and exclusion criteria are summarised in Table 1.

## Findings

### Search results

As shown in Figure 1, a total of 471 papers were initially identified through PubMed ($n = 336$), EbscoHost ($n = 64$), Google Scholar ($n = 38$), results of other reviews ($n = 24$) and peer reviewers ($n = 9$). After duplicates were removed, 204 titles and abstracts were screened, where 100 records were excluded based on the exclusion criteria. Two papers were excluded due to language, one was in Turkish, the other in Portuguese. Following a screening of 104 full-text papers, an additional 67 papers were excluded due to not having a primary focus on SMI, no clear focus on recovery, and no apparent linkages with sectors other than health. This resulted in 37 papers being included for qualitative synthesis (Table 2).

### Included studies

An overview of included studies is presented in Table 2. Studies from an array of countries were included: Bosnia and Herzegovina ($n = 1$); Egypt ($n = 1$); Eswatini ($n = 1$); Ethiopia ($n = 1$); Ghana ($n = 1$); Indonesia ($n = 1$); Kenya ($n = 1$); Kyrgyz Republic ($n = 1$); Liberia ($n = 1$); Nepal ($n = 1$); Chile ($n = 1$); Timor-Leste ($n = 1$); Turkey ($n = 2$); Brazil ($n = 3$); China ($n = 4$); South Africa ($n = 4$) and India ($n = 12$). A variety of study designs and methodologies were reported, including various qualitative studies, randomised control trials, desktop and secondary data reviews, quasi-experimental studies and case studies.

### Overview of SMI recovery approaches

Several different approaches to supporting recovery were highlighted. A common initiative was the establishment of community-based psychosocial rehabilitation centres, which were often run as a collaborative between family members, mental health professionals, and other community resources, to provide psychosocial, job and basic needs support to people with SMI, described in India, Turkey and the Kyrgyz Republic (Molchanova, 2014; Soygür et al., 2017; Kallivayalil and Sudhakar, 2018; Pfizer and Kavitha, 2018; Saha et al., 2020). Also, the clubhouse model for psychosocial rehabilitation was reported in China (Chen et al., 2020), one case describing linkages with a supported employment programme in surrounding communities (World Health Organization, 2021). Supported housing, especially focusing on those experiencing poverty and homelessness, was described in India, (Anish, 2013; Padmakar et al., 2020; World Health Organization, 2021) and Brazil (Acebal et al., 2021). In some instances, mental health teams performed various services, for instance facilitating residential training and placement according to individual preferences and needs – an example is a Recovery Oriented Services (ROSeS) team in India facilitating placement at a rural development centre for an individual who was interested in agriculture and animal husbandry (Vijayan, 2021). Some teams, for instance, a mental health outreach team in India, also facilitated service access through telepsychiatry (Rao et al., 2022), and others, like Atmiyata, facilitated access to government-based social benefits including pensions, rural employment grants, disability benefits and other financial assistance, through the establishment of mental health champions (World Health Organization, 2021).

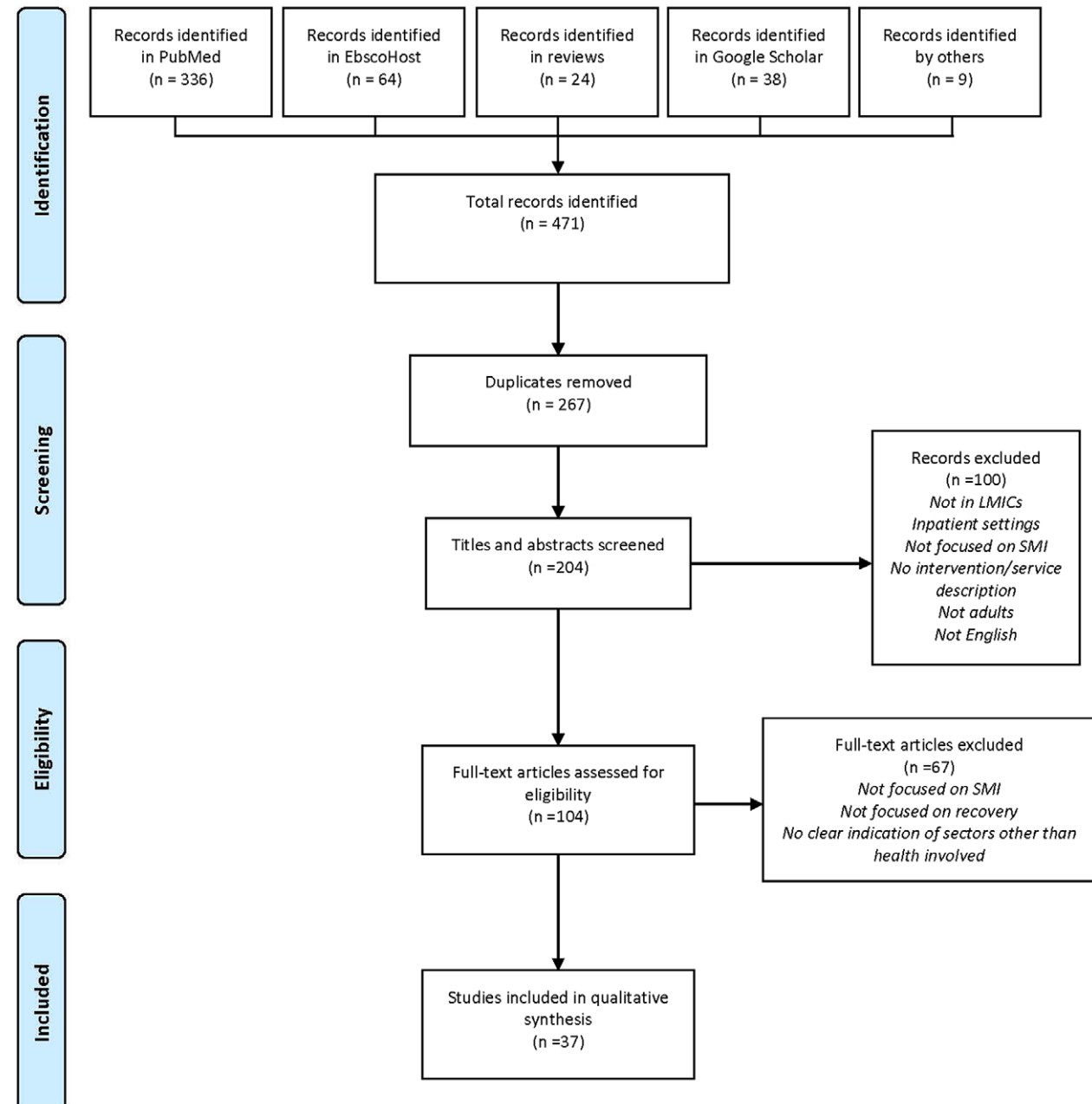

**Figure 1.** PRISMA illustration of search and selection process.

Recovery models that included task-sharing of services to non-specialist workers were reported in South Africa (Brooke-Sumner et al., 2018) and India (Chatterjee et al., 2014). A Critical Time Intervention with task-sharing (CTI-TS) was reported in Chile and Brazil, involving psychosocial support during the transition from psychiatric hospital discharge to community settings (Mascayano et al., 2022). A case describing a multifamily group intervention based on trialogue, psychosis seminars, and co-learning was described in Bosnia and Herzegovina (Muhić et al., 2022), with an NGO-delivered multicomponent intervention for people with SMI and caregivers that included biomedical treatment and supporting economic independence in Nepal (Raja et al., 2012). The salience of integration with religious practices was described in Java (Subandi, 2015) and Egypt (Rashed, 2015). In China, a national pilot pro-gramme was described that involved psychosocial rehabilitation

through cooperation among government organisations and between government and other relevant social organisations (Li and Ma, 2021). A study in Ethiopia seeking to develop a community-based rehabilitation intervention for people with schizophrenia, focused on the development of a specific cadre of worker that would facilitate better networking with other NGO services and expand to other forms of disability as well (Asher et al., 2015; 2022).

### Dimensions of multi- and intersectoral collaboration in supporting recovery

As suggested by the number of papers included in this synthesis, very few examples could be found that explicitly highlight the involvement of sectors other than health in recovery processes in community settings. Only one study described an intersectoral

**Table 2.** Overview of included studies

| | Citation | Country | Primary intervention | Dimensions of involvement of different sectors in recovery | Study type | Data type |
|---|---|---|---|---|---|---|
| 1. | Acebal et al. (2021) | Brazil | Psychosocial rehabilitation | Community-based Residential Therapeutic Services (SRT) in conjunction with routine medical care | Cross-sectional survey | Quantitative |
| 2. | Anish (2013) | India | Psychosocial rehabilitation | Collaboration between healthcare and agriculture industries to promote recovery through vocational activities in diary and farming, and horticulture | Programme evaluation | Quantitative |
| 3. | Arahanthabailu et al. (2022) | India | Assertive community treatment | Manipal Assertive Community Treatment (M-ACT) teams liaise with community resources to facilitate vocational rehabilitation and access to welfare benefits | Quasi-experimental | Quantitative |
| 4. | Arias et al. (2016) | Ghana | Prayer camps | Collaboration between biomedical services and faith-based care delivered at prayer camps | Exploratory qualitative study | Qualitative |
| 5. | Asher et al. (2015), Asher et al. (2022) | Ethiopia | Community-Based Rehabilitation Intervention for People with Schizophrenia | Using community-based rehabilitation workers as principal deliverers of the intervention allows networking and integration with NGOs and traditional health practitioners | RCT | Quantitative and qualitative |
| 6. | Brooke-Sumner et al. (2016) | South Africa | Intersectoral psychosocial rehabilitation | Despite very little formal collaboration between government departments, singular examples emerged, for example, collaboration between social development and public works in placing people living with schizophrenia in an employment programme | Exploratory qualitative study | Qualitative |
| 7. | Brooke-Sumner et al. (2018) | South Africa | Psychosocial rehabilitation (group-based, psychoeducation) | Psychosocial rehabilitation programme delivered by auxiliary social workers via collaboration between the health and social development/welfare sectors | Quasi-experimental | Mixed methods |
| 8. | Chatterjee et al. (2014) | India | Collaborative community-based care | Collaborative package of community-based care facilitated linkages between service users and (1) user-led support structures, (2) community support agencies to address social issues and improve social inclusion and (3) community agencies that provide legal and employment services | RCT | Quantitative |
| 9. | Chen et al. (2020) | China | Psychosocial rehabilitation (clubhouse model) | The Clubhouse facilitates transitional employment and supported education programmes in collaboration with community partners | RCT | Quantitative |
| 10. | de Menil et al. (2015) | Kenya | Mental Health and Development model | Collaboration between the Ministry of Health, NGOs, the Ministry of Gender, Children and Social Services, and Ministry of Agriculture, Livestock and Fisheries, to provide a continuum of services including self-help groups and training in and support for livelihood and farming capacities | Cost-effectiveness analysis | Primary and secondary quantitative data |
| 11. | Gamieldien et al. (2022) | South Africa | Perceptions on recovery | NGOs provide multisectoral services to aid recovery efforts, including basic needs, transportation, life skills, vocational training and leisure and sport activities | Exploratory qualitative study | Qualitative |
| 12. | Hall et al. (2019) | Timor-Leste | Intersectoral mental health service collaboration | Referral linkages between government health facilities, police, local authorities, private clinics, social sector service providers, and | Case study | Mixed methods |

*(Continued)*

**Table 2.** (*Continued*)

| Citation | Country | Primary intervention | Dimensions of involvement of different sectors in recovery | Study type | Data type |
|---|---|---|---|---|---|
| | | | customary healers, forming a network of services that include health care, disability support, victimisation support and residential support | | |
| 13. İncedere and Yildiz (2019) | Turkey | Case Management for Individuals with Severe Mental Illness | A case manager liaises with employment sector to identify suitable candidates to undertake an examination and be positioned for an appropriate job | Quasi-experimental | Quantitative |
| 14. Janse Van Rensburg et al. (2018) | South Africa | Service referrals between state and non-state actors | Government clinics and hospitals refer service users to NGOs for residential support and welfare grant application assistance | Case study | Mixed methods |
| 15. Kallivayalil and Sudhakar (2018) | India | Low-cost community psychosocial rehabilitation model | NGO has a candle-making unit where people living with severe mental illness can sell candles at nearby churches during holy days, while clothes manufacturing initiative linked service users up with retail outlets to sell clothes | Quasi-experimental | Survey |
| 16. Kohrt et al. (2015) | Liberia | Crisis Intervention Team (CIT) Model of Police–Mental Health Collaboration | The Carter Center Mental Health Program (TCC-MHP) facilitated partnerships to advance mental health policy, legislation, and funding, which included engaging with the Liberia National Police to identify spaces for collaboration on crisis intervention | Programme development description | Narrative description |
| 17. Li and Ma (2021) | China | National comprehensive management pilot project for integrated care for people with severe mental disorders through strengthened cooperation among government organisations and between government and other relevant social organisations | A quasi-governmental organisation establishes and coordinates community service organisations for people living with severe mental illness, with formal links between the ministries of health and social affairs. Through training and cooperation across a range of organisations and sectors, an integrated package of services is provided to be more responsive to individual needs | Case study | Qualitative interviews |
| 18. MacDougall et al. (2022) | Kenya | Community REcovery Achieved Through Entrepreneurship (CREATE) | An initiative that integrates elements of psychosocial rehabilitation (PSR), community-based rehabilitation (CBR), and work integration social enterprise (WISE) | Programme development description | Qualitative |
| 19. Mascayano et al. (2019), (2022) | Brazil and Chile | Critical Time Intervention with Task-sharing (CTI-TS) | Teams made up of auxiliary and peer workers supported service users following discharge from acute psychiatric hospitalisation, to facilitate linkages with a range of community-based support systems that included basic and specialist medical care, psychosocial rehabilitation, leisure and art-based activities and basic needs | RCT | Quantitative |
| 20. Molchanova (2014) | Kyrgyz Republic | Indigenous model of family rehabilitation | The development of family-driven NGOs led to the provisioning of a range of psychosocial rehabilitation activities to people living with severe mental illness in community settings | Desktop review | Government documents |
| 21. Muhić et al. (2022) | Bosnia and Herzegovina | Brief, multifamily group intervention for patients with schizophrenia and related disorders | Multifamily groups mobilised mutual support for people living with severe mental illness in community settings | RCT | Survey and qualitative interviews |

**Table 2.** (*Continued*)

| Citation | Country | Primary intervention | Dimensions of involvement of different sectors in recovery | Study type | Data type |
|---|---|---|---|---|---|
| 22. Nxumalo Ngubane et al. (2019) | Eswatini | Psychiatric outpatient care | People living with severe mental illness' engagement in community-based projects such as community kitchens for orphaned and vulnerable children aided in recovery efforts | Interpretive phenomenological analysis | Qualitative interviews |
| 23. Padmakar et al. (2020) | India | The Banyan's supported housing model | The Banyan organisation developed a supported housing programme where people living with severe mental illness can live independently, with an emergency care and recovery unit located in close proximity | Mixed methods | Qualitative interviews, logbook notes, survey |
| 24. Pfizer and Kavitha (2018) | India | Interdisciplinary recovery model of psychosocial rehabilitation | People living with severe mental illness were enrolled into a sheltered workshop, where their functionality, occupational skills and readiness were improved and evaluated, after which appropriate service users could be promoted to peer mentorship at a trial worksite, with the ultimate goal of securing competitive employment. The programme also included generating financial support for the building of houses on private owners' properties, as well as linkages with Alcoholics Anonymous to address substance abuse challenges | Desktop review | Narrative description |
| 25. Raja et al. (2012) | Nepal | BasicNeeds model of Mental Health and Development | Livelihood support was provided to people living with severe mental illness and their families through cash grants or supporting the setting up of businesses, as well as support for the setting up of self-help groups | Case study | Project data |
| 26. Rao et al. (2022) | India | SCARF Telepsychiatry in Pudukkottai (STEP) program | A range of community-based psychosocial rehabilitation activities was provided in a rural area, including facilitating access to disability and welfare benefits, supporting job-seeking efforts and facilitating placement in partner businesses, and supporting the obtaining of loans from banks to help set up small businesses | Desktop review | Project data |
| 27. Rashed (2015) | Egypt | Qur'anic healing | A duality of recovery care that consisted of psychiatric services delivered by medical doctors, and Qur'anic healing providing spiritual care | Ethnography | Participant observation |
| 28. Saha et al. (2020) | India | Non-governmental psychosocial rehabilitation centres | NGO that provided support for livelihood activities, access to government grant schemes, as well as a range of psychosocial therapies with service users and their families | Secondary data analysis | Patient case records |
| 29. Soygür et al. (2017) | Turkey | Therapeutic community and supported-employment setting where people living with schizophrenia work | Blue Horse Café provides a protective space where people living with schizophrenia can work, build skills and develop independence from medical institutionalisation, while still accessing medical care | Phenomenological | Qualitative interviews |
| 30. Subandi (2015) | Indonesia | Psychiatric outpatient care | In addition to outpatient community-based medical care, people accessed "natural therapy" in the form of spiritual guidance from mosques alongside neighbours and friends, which also allowed for community integration, while others accessed | Ethnography | Participant observations |

(*Continued*)

**Table 2.** (Continued)

| Citation | Country | Primary intervention | Dimensions of involvement of different sectors in recovery | Study type | Data type |
|---|---|---|---|---|---|
| | | | services from both biomedical and traditional health practitioners | | |
| 31. Vijayan (2021) | India | Recovery Oriented Services (ROSeS) | Community-based psychiatric rehabilitation programme aids in recovery efforts by acting as an intermediary between service users and organisations that facilitate work placements | Case study | Qualitative |
| 32. World Health Organization (2021) | Brazil | Centro de Atenção Psicosocial (CAPS) | Drives recovery efforts by facilitating active citizenship, which includes helping service users to navigate government bureaucracies to obtain formal documentation and access benefits, and liaising with a range of community resources to support housing, employment and social life improvement | Case study | Narrative description |
| 33. World Health Organization (2021) | China | Phoenix Clubhouse | Clubhouse collaborates with business partners to facilitate placement for paid employment in the local labour market for its members | Case study | Narrative description |
| 34. World Health Organization (2021) | India | Naya Daur Community Outreach | A community outreach programme that refers people to temporary shelters, where they can access basic services such as hygiene materials, food, water and a place to sleep | Case study | Narrative description |
| 35. World Health Organization (2021) | India | Atmiyatab primary care community outreach service | Mental health champions are instituted as intermediaries for people living with severe mental illness and their families to access disability certification, to access government benefits such as pensions, grants and disability benefits, as well as work schemes | Case study | Narrative description |
| 36. World Health Organization (2021) | Georgia | Hand in Hand supported living | An organisation providing employment support by collaborating with community social enterprises and employers | Case study | Narrative description |
| 37. World Health Organization (2021) | India | Home Again housing and supportive services | The Banyan organisation's housing and supportive services initiative provides access to housing, establishing work placements, educational support, and linking with various community resources to promote recovery efforts | Case study | Narrative description |

collaboration between health and other sectors in supporting SMI recovery on a national, policy-level scale, describing the formalising of governance and funding structures for better interorganisational collaboration and funding in China (Li and Ma, 2021). In terms of programmatic interventions, several dimensions of multi- and intersectoral collaboration emerged, described below in terms of Health and Housing, Health and Community Support Systems, Supportive Community Spaces for Recovery, and Bridging Biomedical and Social Spheres of Care through Lay Health Workers.

### Health and housing

There were instances of collaboration between the health sector and various actors involved in providing supported housing to people living with SMIs. The Phoenix Clubhouse in Hong Kong, China,

put in place arrangements with housing partners, including public housing, supported hostels, halfway houses, long-stay care homes and residential respite services, which members can access (World Health Organization, 2021). An Indian study (Anish, 2013) reported that the majority of residential facilities for people with SMI were provided by faith-based organisations with funds from public donations. These faith-based organisations tended to collaborate with other sectors during the period when service users are admitted to the centre following referral by mental health professionals, family members, police and social services.

An example of this kind of collaboration is the Banyan's supported housing model. The Banyan started out as a crisis intervention and rehabilitation centre for homeless women with mental illness in the city of Chennai, India, and has expanded its services to

include emergency, open shelter and street-based services, social care and long-term and alternative living. In support of living arrangements, the organisation entered into rental agreements with private property owners in order to secure housing for people with SMI, who were supported through stages of confrontation, adaptation and stabilisation (Padmakar et al., 2020). A Brazilian study (Acebal et al., 2021) investigated service users' perspectives on the relationship between housing needs and mental health/illness. It highlighted the importance of the links between "residential therapeutic services" (supported housing) and biomedical health facilities but details of the working relationships between health facilities and residential facilities were lacking.

### Health and community support systems

A key area for multi- and intersectoral collaboration is the setting up and strengthening of community-based support resources beyond the health sector. In the aforementioned CTI-TS model in South America, lay community mental health workers and peer support workers formed CTI teams that provided structured, time-limited support to people discharged from psychiatric hospitalisation. Working from community mental health centres, a key task in this initiative was to support beneficiaries through linking them to informal and formal support systems in communities (including local leisure clubs and community centres) after which a gradual withdrawal period would take place thereby lessening dependence on the CTI programme or institutional mental health services (Silva et al., 2017; Mascayano et al., 2019; Mascayano et al., 2022). The multifamily support group model in Bosnia and Herzegovina served to mobilise mutual support in community settings (Muhić et al., 2022), Two studies described mental health service networks across sectors, that included support for people living with SMI. In Liberia, The Carter Center Mental Health Program (TCC-MHP) partnered with the Liberian police sector to develop Crisis Intervention Teams (CIT) to create more supportive services for people living with SMI (Kohrt et al., 2015). In Timor-Leste, collaboration and referral between mental health and social service delivery platforms were reported, that included referral from police, local authorities, private care and social services to government health facilities for care, particularly those living with SMI. Government services in turn referred people to support organisations, including housing support for people living with SMI (Hall et al., 2019). In a similar study from South Africa, a range of NGO activities were described, where people living with SMI were sometimes referred to organisations for housing and basic needs support, as well as to a social services organisation that provided home-based psychotherapy, group therapy, social support, community awareness and education campaigns. There were also instances of collaborating with old-age facilities to provide housing support to people with SMI (Janse Van Rensburg et al., 2018).

### Supportive community spaces for recovery

Given the history and prevalence of stigma, discrimination and structural barriers to social integration faced by people living with SMI, recovery processes require safe and supportive spaces in communities. The Centro de Atenção Psicosocial (CAPS) in Brazil is a network of community-based mental health centres, which promotes active citizenship through a range of services, including supporting people through the various bureaucracies of obtaining formal documentation and access social support, training and education, access to supportive housing and supportive work placement, with collaborations across the sectors of health, education, justice, social assistance and various non-governmental agencies

(World Health Organization, 2021). The well-known clubhouse model of psychosocial rehabilitation, with its roots in the 1940s in New York, was applied in Chinese settings and involved non-residential services that included employment and supported education programmes, linked with private and education sectors (Chen et al., 2020; World Health Organization, 2021). Another example is the Blue Horse Café in Turkey, a therapeutic community and supported-employment setting where most services offered by the café are performed by people living with SMI, including food preparation and serving, reservation management, cleaning, management and organisation and selling of second-hand goods. This provides a protective environment within which people with SMI can participate in the labour sector, while also receiving therapeutic support (Soygür et al., 2017). Another programme in Turkey assisted service users through case management, where people were supported in preparing CVs and job interviewing, interviews with labour agencies, and reviewing of vacancies. People were also accompanied during job interviews, and during their first days of employment, and relationships were established between case managers and line managers in workplaces (İncedere and Yildiz, 2019). In India, the Rajah Rehabilitation Centre (RRC) collaborates with employers to secure employment for people with SMI, following a period of supported work and skills development. There is also collaboration with community-based Alcoholics Anonymous and Al-Anon support groups to support participants and their families who have to deal with challenges related to substance abuse, while legal services are made accessible through a collaboration with a Legal Aid clinic (Pfizer and Kavitha, 2018). The development of peer support networks in Kenya connected people living with SMI with skill-building in livelihood activities, such as drought-resistant farming and making detergent (de Menil et al., 2015). In Ghana, supportive spaces including housing were provided in prayer camps, overseen by local prophets (Arias et al., 2016).

### Bridging biomedical and social spheres of care through lay health workers

Instances emerged where lay health workers were trained and supervised by mental health professionals to provide community-based services, thereby bridging the domain of healthcare within facilities with the social dimensions of recovery in community settings. In the community-based intervention for people with schizophrenia and their caregivers in India (COPSI), the programme included the linkage of people with SMI with community agencies and user-led self-help groups. This provided a support for seeking employment as well as to access social and legal benefits (Chatterjee et al., 2014). A similar intervention was described in South Africa, where a community-based psychosocial rehabilitation intervention was delivered in partnership with PHC health clinics and a local NGO by auxiliary social workers. Participants for the intervention were recruited through clinics and intervention conducted in clinic premises by auxiliary social workers (Brooke-Sumner et al., 2018). A Nepalese study of an NGO delivered multi-component intervention for people with SMI and caregivers (access to biomedical treatment and enabling service users and caregivers to develop a livelihood) recommended expanding scope of training of community health workers to include skills in delivering support for sustainable livelihood interventions (Raja et al., 2012).

### Barriers to multi- and intersectoral collaboration

Though difficult to assess comprehensively due to the ambiguity of descriptions of multi- and intersectoral collaboration, limited

barriers to such collaboration emerged. A key barrier highlighted in the Chinese interorganisational collaboration case was differences in commitment and professional authority between organisations, both government and non-government. Specifically, there stronger institutional commitment of actors in the health sector was reinforced by the greater degree of professional authority wielded by psychiatrists (Li and Ma, 2021). The Blue Horse Café case in Turkey highlighted contrasts in relationships with healthcare workers versus relationships in a community-based therapeutic community, with descriptions of the former cold, indifferent or lacking in sincerity, whereas the humanistic aspects of the latter were detailed in terms of equal power relations and mutual respect. The space was described as supportive of power-sharing between health workers and people living with SMI (Soygür et al., 2017). However, not all healthcare workers might feel comfortable working outside the spheres of health facilities. In the reporting of the Banyan-supported housing model, healthcare workers experienced challenges adapting to their roles in community settings and social rather than biomedical orientation. There was also a cultural dimension, in that unmarried female healthcare workers felt pressure to justify them living unmarried in the community where they worked (Padmakar et al., 2020). In the Chinese case, a lack of role clarification for frontline workers attending to multiple vulnerable populations and working across sectors resulted in them experiencing increased pressure to deal with SMI. Also, the dominance of the Chinese government resulted in cooperation between government and social organisations being driven by the willingness of government organisations to work with social organisations (and not vice versa), thereby skewing the power differential towards government departments rooted in psychiatric professional expertise (Li and Ma, 2021). Nonetheless, intersectoral working was codified in formal arrangements, which is not the case in many other settings. In South Africa, there is a recognised need for input from social services, education and labour into recovery programmes (Gamieldien et al., 2022). Further, recommendations were made that the Department of Health, Department of Social Development and NGO sectors should improve communication between sectors, promote leadership from all levels and formalise intersectoral relationships through appropriate written agreements (Brooke-Sumner et al., 2016). The lack of formal agreements and intersectoral policy was also highlighted in Timor-Leste, which often translated into limited prioritising of mental healthcare (Hall et al., 2019). Finally, there are persistent structural barriers faced by organisations and individuals alike when pursuing SMI recovery in community settings. For instance, the Banyan model faced challenged from private property owners when attempting to secure housing for their beneficiaries, which included stigmatising attitudes towards people living with SMI and enacting a preference for residents who are more functional and mobile (Padmakar et al., 2020).

## Discussion

The aim of this scoping review was to identify and describe multi- and intersectoral approaches to enable SMI recovery in LMICs. The principal finding of this scoping review is that while such approaches have been widely supported in literature on developing appropriate and responsive support systems for SMI, very few studies operationalise and describe how multi-and intersectoral work is done in relation to recovery services in LMICs. This is in contrast to the comparatively vast body of work on multisectoral

interorganizational collaboration and networks for community SMI care developed in HICs (Rosenheck et al., 1998; Fleury and Mercier, 2002; Morrissey et al., 2002; Wiktorowicz et al., 2010; Whiteford et al., 2014; Lorant et al., 2017; Jørgensen et al., 2020; Nicaise et al., 2021). From its origins from a WHO technical working group who realised that optimal sanitation requires a coordination between traditional public health and infectious disease actors and engineering and water management specialists (de Leeuw, 2022), intersectoral action has gained traction in global health discourse, though this has not been robustly translated to services supporting SMI recovery. Our review highlights several areas where multi-and intersectoral collaboration has been demonstrated in LMICs with respect to community support for people living with SMI, particularly in the areas of housing support, the development and sustainment of protective spaces where recovery can take place, and linkages between health facilities and community resources.

The Integrated Recovery Model posits that each individual has subjective recovery needs, centred around basic needs such as accommodation and employment, as well as less tangible needs such as coping skills and hope. Three core components interact with these needs: remediation of functioning (recovering mental and physical well-being), collaborative restoration of skills and competencies (building hope through collaborative restoration of agency, function and participation), and active community reconnection (re-establishing a place in the community with a range of skills and supports). Importantly, these processes unfold in linear and overlapping fashion. During the critical period following deterioration of well-being, remediation comes into play, where collaboration between various community actors and sectors and acute mental health services are critical. During the restoration period, psychosocial rehabilitation becomes central, which again necessitates the mobilisation of collaborations and resources in LMIC settings. Finally, moving towards an achievement of a degree of recovery, various community-based actors including NGOs, faith-based actors and community members becomes key (Frost et al., 2017). Our findings here suggest that, in most settings, elements of this model can be found, addressing the key movements from psychiatric relapse to recovery and community integration – whether through initial referral for specialist care, fulfilling basic needs or sustaining safe spaces and collaborations across sectors for people to recover.

Importantly, in many societies where tightly-knit families and high level of social cohesion are prevalent, especially in African and sub-Indian continental communities, the family has (and continues to be) a central locus of care beyond the boundaries of facility-based mental healthcare (Alem et al., 2008; Chadda, 2012). Family caregivers in LMIC are key to creating an environment that supports recovery but the burden of care is compounded by less-developed community systems of care, marked social stigma and certain cultural practices (Karambelas et al., 2022). People with SMI and their caregivers face barriers to securing formal income or employment, food, housing, transport and education (Addo et al., 2018). Holistic care for this vulnerable group is thus intricately linked with poverty alleviation, development and working towards social inclusion (Lund et al., 2011; Jenkins et al., 2011a, 2011b, 2011c; Plager-son, 2015) all of which have been hampered by the impact of COVID-19 (Kola et al., 2021).

Several studies indicate the leading coordinating role of non-governmental or charitable organisations (e.g., BasicNeeds) in bringing together stakeholders from other sectors for recovery-focused work. While this may be effective, NGOs are commonly

reliant on donor funding and programmes may not be sustained in the long term and the corresponding influence in coordinating intersectoral action may be eroded. While a whole of government approach is indicated it is likely that one stakeholder or partner sector should take a leading coordinating role in sustaining intersectoral work and this need not be the health sector. In principle, this involves moving away from a purely biomedical model of treatment and recovery for SMI in which sectors other than the health sector recognise the role of the social environment in creating psychosocial disability associated with these conditions (World Health Organization, 2019). The benefits of such a shift also include sharing of the economic burden of SMI across sectors, a critical step away from health facility-focused spending (OECD, 2021). Although the literature on operationalised intersectoral and multisectoral work for recovery is limited, findings of this review suggest overarching domains for action that may be pursued (in context-specific ways) to drive intersectoral work in LMIC. These are: (1) building relationships between key actors including people with lived experience and families; (2) prioritising supportive spaces for recovery that help with fulfilling basic needs; (3) building leadership capacity among actors to solidify and formalise intersectoral work and (4) integrating resource allocation between actors to underpin these approaches. Country-specific approaches to may also benefit from leapfrogging, that is, harnessing strategies previously used in intersectoral initiatives of disability movements and advocacy for treatment and care for HIV and TB in LMIC.

## Limitations

The main limitation of this scoping review was the exclusion of non-English language papers, and of grey literature. This approach was taken as a feasible first approach to scoping the literature on this topic. A future review should consider inclusion of grey literature, given the importance of the non-profit/non-government sector in provision of community-based services for recovery. This review also did not include searches specifically looking at social welfare payments and health insurance coverage for treatment costs (which are available in some LMIC and may be considered a form of intersectoral work). A further scoping review is in process that will cover this topic.

## Conclusion

Multi- and intersectoral collaboration lies at the heart of recovery – "medical solutions to social problems are expensive, ineffective and inefficient," and integration between the biomedical and the social is "humane, cost-effective and truly recovery-oriented" (Drake and Whitley, 2014). In this review, we have described limited, though promising, examples of such action. This hopefully serves as a call for researchers, policymakers and service providers to both work more deliberately with other sectors and to strive to be more strategic in doing so, while keeping the recovery needs of the individual at the centre of actions.

## Considerations for the future

As the barriers outlined in the findings suggest, these are often piecemeal and relatively uncoordinated ventures, and require more deliberate, strategically coordinated actions. Multi- and intersectoral action for health and well-being involve strategies and action plans; long-term multisectoral and intersectoral initiatives; permanent structures; projects; legislative or parliamentary decisions and tools (World Health Organization, 2018). The following five recommendations have been suggested to facilitate intersectoral action for mental health in LMICs (Skeen et al., 2010) which align with the four domains for action described above:

1. Develop supportive legislation and policy alongside the other formalised structures for intersectoral action
2. Develop leadership in the health sector and beyond, especially in cross-cutting agencies
3. Employ targeted awareness-raising to engage all relevant sectors in order to specify roles, responsibilities and strategies
4. Develop a formal, structured approach to intersectoral action for mental health to address the lack of dedicated budgeting and unclear roles
5. Drive intersectoral work on a microlevel, in order to effectively address basic services such as water, electricity and sanitation

The principles for recovery-related service delivery should further be couched in these structures, including that services are person-centred, holistic and inclusive; enable agency and self-management; integrated across the care continuum; seamless and complementary across government departments, NGOs and other services; evidence-based; underlines equity in choosing service options; and are aligned with national, national and local strategy (Frost et al., 2017).

In terms of the findings reported here, generating universal recommendations or actions is challenging given the wide array of health and social systems across countries and regions. Nonetheless, there are thematic clusters that could provide direction to policymakers and other stakeholders in strengthening recovery efforts in an inter and multisectoral way. A common strategy that emerged relates to providing housing support, which ranged from communal, semi-institutionalised recovery settings to independent living in supportive housing arrangements. This requires the acquisition and appropriate management of physical spaces and require partnerships between the health sector, public civil infrastructure sector as well as private citizens and organisations, and formal arrangements through public–private partnerships should be set in place. Regarding community support systems, many examples here highlight linkages between people living with SMI and their caregivers, and various community-based resources. This requires a community-based body, organisation, clinic or government agency (depending on the health system configuration) that can set up relationships with and curate a list of a range of resources that people can be referred to. Especially in lower-resource settings, it is essential to tap into existing resources beyond the health sector that often remain underutilised in supporting people living with SMI. Lessons can be gleaned from many examples of multisectoral collaboration in addressing HIV, TB and non-communicable diseases. In terms of supportive spaces for recovery, many NGOs provide such environments with various types and degrees of support, much of which relates to assistance in navigating the bureaucracies involved in accessing grant schemes as well as supporting people to access the job market. This requires willing employers and a supportive employment work environment, which, given perpetuating stigma, would require educational investment as well as buy-in from appropriate employers. In terms of the bureaucracies of accessing grant schemes, more can be done by government agencies to remove administrative obstacles for people living with SMI, for example, making the medical diagnostic screening process more accessible. Finally, it is crucial to develop an

appropriate health worker mix to deliver the complex range of activities within the ambit of recovery and inter- and multisectoral approaches. Task-sharing and empowering lay health workers have grown substantially as a viable option for constrained settings, and lay health workers can potentially offer crucial linkages with sectors and resources outside of the health sector. Nonetheless, these workers need appropriate training, regulation and a supportive career pathway in order to sustain their role in recovery-oriented services.

Finally, a particular challenge that emerged during the search and screen phases of this review was the imprecision evident in descriptions of multi-and intersectoral collaboration. Descriptions of the roles and remits of sectors were lacking, and are required to give context to the application of multi or intersectoral work. Intersectoral collaboration has been described as "an intricate web of interdependent organisations, individuals and behaviours, implicitly or explicitly driven by beliefs or assumptions to pursue a set of interconnected ideals, goals and objectives through the variously dispersed and joint control and allocation of resources" (de Leeuw, 2022). Given this complexity, an approach to render descriptions of multi- and intersectoral work more explicit could be for reviewers and journals involved in publishing recovery-based studies and interventions to request details on the ways that partnerships are formed and maintained (partnership working as a heading).

**Open peer review.** To view the open peer review materials for this article, please visit http://doi.org/10.1017/gmh.2023.10.

**Supplementary material.** To view supplementary materials for this article, please visit https://doi.org/10.1017/gmh.2023.10.

**Data availability statement.** Data are available on request from the corresponding author.

**Author contributions.** A.J.R. and C.B.-S. conceptualised the study, conducted the search, screening and analysis and co-wrote and endorsed the final draft.

**Financial support.** A.J.R. receives funding from the National Institute for Health Research (NIHR) (NIHR201816) using UK aid from the UK Government to support global health research. The views expressed in this publication are those of the author and not necessarily those of the NIHR or the UK Department of Health and Social Care. Research reported in this publication is supported by the National Institute of Mental Health (NIH) under award number U19MH113191-01 and 1R34MH131233-01. The content is solely the responsibility of the authors and does not necessarily represent the official views of the National Institutes of Health. This work is based on the research supported in part by the National Research Foundation of South Africa. C.B.-S. is supported by grants from the South African Medical Research Council with funds received from the South African National Department of Health and the UK Medical Research Council, with funds received from the UK Government's Newton Fund.

**Competing interest.** The authors declare no conflict of interest.

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
