## [Reviewer Report]

*Comments to Author*: Overall, this is a well-written review that addresses a neglected area. The main concern is whether important studies have been missed due to the search strategy.

Abstract

1. Consider deleting this sentence: “Importantly, there is a need to explore the meaning of the concepts of recovery, multi and -intersectoral approaches in terms of LMIC contexts.” It is not clear that this review does this (it is not part of the aim to explore meaning of concepts of recovery).

2. Please include some methodological detail about databases included, the time period covered, inclusion/exclusion criteria, etc.

3. Please indicate the number of papers included, whether they come from LICs/UMICs and/or geographical spread, the nature of the study designs and focus.

4. The conclusion reads as findings, aside from the last sentence.

Introduction

5. Sentence 1 – in low-income countries, undernutrition and communicable diseases remain the leading causes of premature mortality in people with severe mental illness, e.g. Fekadu et al. British Journal of Psychiatry, although few studies have been conducted in these settings.

6. The Gamieldien scoping review problematises the fit of Western concepts of recovery to non-Western and LMIC settings (while also identifying similarities). It would strengthen this introductory paragraph to recognise that.

7. There have been efforts to adapt ICCCF to incorporate multi-sectoral action, e.g. Mall et al. Epi and Psych Sciences, 2016 (rural Ethiopia). But it is true that this has not been picked up and brought into the mainstream.

8. The US deinstitutionalisation example is good to show consequences of failure to have interdisciplinary structures. But are there examples of successful application of this principle in HICs? Here is the place to include that. Otherwise, the reader is led to ask why you have only focused on LMICs in this review. Indeed, in general, better elaboration of the rationale for focusing on LMICs is needed (see recent critiques of LMIC-focused reviews in BMJ Global Health e.g. Lencucha and Neupane).

Methods

9. How did you search google scholar? Please be more specific e.g. did you restrict to the first set of pages? Or only references with pdfs? Because otherwise the number of hits is usually unmanageable.

10. What does this mean? “Multiple combinations were used to yield a workable batch of papers to include.” This does not sound like a systematic approach to reviewing. The number of records identified (n=454) seems smaller than expected.

11. Why restrict medline to ‘full text’? That unnecessarily restricts the search and can introduce bias.

12. Describe in the main paper how you operationalised ‘SMI’ (which kinds of disorders were in/out), ‘LMICs’ (not just referring to the search strategy).

13. What was the time period for the review? (how far did you go back? When was the end date of the review?)

14. What were the inclusion/exclusion criteria?

15. Who did the screening? Was there any independent screening? How were differences resolved?

16. Please include the details on number of papers identified in the findings, not the methods.

17. Please specify separately how many papers were excluded because they were not in English. This is an important limitation and the reader needs to know how biased the included set of papers might be.

Findings

18. Is it possible to include an abbreviated table describing key components of the studies and the service/programme components (explicitly highlighting the inter-/multi-disciplinary aspects) within the main manuscript? Most readers will not easily access appendices and this information is critical to understanding the findings. It is usually within the main manuscript.

19. Singapore is a (very) high-income country and should not be included.

20. A trial of community-based rehabilitation in Ethiopia is missing – Asher et al. Lancet Global Health, 2022 (published May 2022).

21. The Liberia study of developing a collaborative model between mental health services and police also seems relevant to this review – Kohrt et al.

22. The basic needs model in Kenya also seems relevant: https://www.researchgate.net/publication/277080358_Cost-effectiveness_of_the_Mental_Health_and_Development_model_for_schizophrenia-spectrum:and_bipolar_disorders_in_rural_Kenya

23. Were there any examples from the case studies in the WHO 2021 publication examining recovery-oriented community mental health care?

24. This is not clear: “…only point of these faith-based organisations working jointly with other sectors was the time point of admission to the centre” (re. Banyan?).

25. In the description of the blue horse café, I think the second ‘former’ should be a ‘latter’.

26. How much are people with lived experience of severe mental illness involved in the planning of recovery-oriented care?

Discussion

27. This is generally very well written, but please include a limitations section. E.g. around language restrictions, a certain ‘pragmatism’ to make the review manageable, missing grey literature (no systematic search – recent CBR grey lit review shows the relevance in this area of study).

28. The last paragraph is difficult to follow.

Overall

29. There are a few typos – please proof-read carefully.

---

## [Reviewer Report]

*Comments to Author*: This is a well researched and written paper which highlights a key gap in knowledge in relation to inter and multi-sectoral coordination of care for people with serious mental disorders.

A few suggestions to improve the paper:

Cite the recently launched World Mental Health Report which highlights intersectoral coordination in the Services chapter to highlight the relevance of the paper.

Provide a brief summary of examples of inter-sectoral coordination efforts from HICs, as this might be relevant for a readership not familiar with this literature.

In the discussion section, can the authors draw upon the four dimensions of inter-sectoral collaboration to make the recommendations less generic and also identify a set of key actions to promote such coordination in terms of services (like housing, employment, social benefits, social connectedness etc)?

---

## [Reviewer Report]

*Comments to Author*: The authors have comprehensively addressed my comments. The review is now more substantive and makes an important contribution to the field.

---

## [Reviewer Report]

*Comments to Author*: The authors need to be congratulated for highlighting the important and relatively less explored issue of inter-sectoral collaboration as a key structural or systemic dimension of recovery.

The principal findings reflect that, while there are promising examples of such initiatives in LMICs, there is a lack of a coordinated response and an effective mechanism to do so.

Specific comments:

1. The Introduction has this sentence- ‘the social environment for people with SMI, particularly in low and middle-income countries (LMIC), is characterised by poverty and social exclusion which hampers their recovery’. This is somewhat of a sweeping generalization and the situation is far more nuanced. Could the authors consider modifying the sentence to reflect the diversity of outcomes for people with SMIs in LMICs?

2. Based on their findings, could the authors speculate on what are some of the key domains (like participation of people with lived experiences and their caregivers, leadership and governance, financing) of effective inter-sectoral collaboration to promote the recovery of people with SMIs that are feasible in LMICs?

3. In some LMICs, there is access to disability and other social benefits and insurance coverage for treatment costs - are these possible examples of inter-sectoral collaborations that could be included in the paper?

4. Do the authors think that exploration of inter-sectoral initiatives for other disabilities and chronic diseases like HIV/AIDs, TB, NCDs in LMICs (some of which have had considerable success) can have lessons for developing similar collaborations for people living with SMIs?